# Estimating the Impact of Biodiversity Loss in a Marine Antarctic Food Web

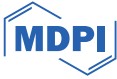

Vanesa Salinas [1,*] , Georgina Cordone [2] , Tomás I. Marina [3] and Fernando R. Momo [1,4]

1 Instituto de Ciencias, Universidad Nacional de General Sarmiento (UNGS), Los Polvorines 1613, Argentina; fmomo@campus.ungs.edu.ar
2 Centro Para el Estudio de Sistemas Marinos (CESIMAR), Centro Nacional Patagónico (CCT CONICET-CENPAT), Puerto Madryn U9120, Argentina; georginacordone@gmail.com
3 Centro Austral de Investigaciones Científicas (CADIC-CONICET), Ushuaia V9410, Argentina; tomasimarina@gmail.com
4 Departamento de Ciencias Básicas, Universidad Nacional de Luján (UNLu), Luján 6700, Argentina
* Correspondence: vsalinas@campus.ungs.edu.ar

**Abstract:** The consequences of climate change and anthropogenic stressors, such as habitat loss and overexploitation, are threatening the subsistence of species and communities across the planet. Therefore, it is crucial that we analyze the impact of environmental perturbations on the diversity, structure and function of ecosystems. In this study, in silico simulations of biodiversity loss were carried out on the marine food web of Caleta Potter (25 de Mayo/King George Island, Antarctica), where global warming has caused critical changes in the abundance and distribution of benthic and pelagic communities over the last 30 years. We performed species removal, considering their degree and trophic level, and including four different thresholds on the occurrence of secondary extinctions. We examined the impact of extinctions on connectance, modularity and stability of the food web. We found different responses for these properties depending on the extinction criteria used, e.g., large increase in modularity and rapid decrease in stability when the most connected and relatively high-trophic-level species were removed. Additionally, we studied the complexity–stability relationship of the food web, and found two regimes: (1) high sensitivity to small perturbations, suggesting that Potter Cove would be locally unstable, and (2) high persistence to long-range perturbations, suggesting global stability of this ecosystem.

**Keywords:** species loss; network properties; species properties; extinction thresholds; complexity; stability

## 1. Introduction

Climate change, together with the impacts of human activities such as habitat fragmentation, pollution, and overexploitation of natural resources, is driving an unprecedented ecological crisis that threatens ecosystems, both terrestrial [1,2] and marine [3,4]. These stressors are changing the patterns of species abundance, distribution, and interactions, moving numerous species towards the brink of extinction, causing alterations at high levels of organization (e.g., food webs) and threatening the persistence of ecological communities [5].

There are many biotic and abiotic factors—ultimately associated with climate change—that can cause species extinction (e.g., changes in temperature, temporal mismatch between interacting species, freshwater scarcity, etc.); however, changes in biotic interactions that lead to a variation in food availability are one of the most evident factors [6–8]. Furthermore, since every species is functionally unique, the risk of a species becoming extinct will also depend on different biological and ecological species traits, such as body size, habitat type and diet breadth [9]. For this reason, it is of great importance to establish and quantify the effects of species extinction on food webs.

Generally, extinction simulations in food webs have been developed considering that a species goes extinct when it loses all its prey due to primary extinctions [10–18]. This assumption does not address the possibility that a consumer may become extinct when a certain percentage of its prey is lost; only a few works have included different extinction thresholds in this sense. For example, Bellingeri and Bodini [19] introduced thresholds as an energetic criterion to define species extinction and analyzed the robustness of ten food webs against random species loss. Cordone et al. [20] simulated ordered and random extinctions in an Antarctic food web, including extinction thresholds, in order to study changes in connectance and the number of secondary extinctions. However, the effects of biodiversity loss on food web properties considering thresholds and different species extinction criteria have been poorly studied.

Food webs comprise the trophic interactions between species in an ecosystem and the flows of matter and energy among organisms; therefore, food webs provide information about key factors that can modify an ecosystem's structure, function and stability [12,21–25]. Thus, it is crucial to understand how food web properties change due to environmental perturbations that cause biodiversity loss [17,26,27], especially in threatened ecosystems, such as polar regions, where climate change is causing warming at a faster rate than other places in the world [28–30]. During the last 30 years, many works have studied the effects of biodiversity loss in food web stability [11,13,14,31–34], embracing this complex concept that contains multiple facets such as resilience, resistance, persistence and robustness [35]. One of the most studied concepts when simulating species loss in food webs is robustness, defined as the fraction of primary species loss that induces at least 50% total species loss [12,13,15,16,20,36–39]. In relation to this concept, many studies have highlighted the high number of secondary extinctions after removing most-connected species, addressing its cascading effect on food webs [11–13,26,40,41]. Recent works have assessed food web robustness by including different network properties [42,43]; however, little is known about how predator sensitivity to prey loss and extinction criteria (e.g., removing species at the trophic level) affect the complexity and stability of food webs.

In this study, we performed in silico extinctions on the highly-resolved food web of the Potter Cove marine ecosystem (25 de Mayo/King George Island, Antarctica). This ecosystem is considered a biodiversity hotspot, where global warming has caused the decline of coastal glaciers and the increase in suspended particles due to ice melting [44–46], which has had profound implications for the benthic and pelagic communities [47–50]. Although several works have focused on quantifying the biological data (e.g., biomass, body size and body mass) of Potter Cove's benthic communities, such as macroalgae [51] and meiofauna [52], there is a big gap in biological information on many other taxa (e.g., mollusks and echinoderms). Also, the different sampling methods (dredges, trawls, photographs, etc.) and parameters (biomass as wet weight or dry weight, densities, percentage cover, etc.) used, makes it difficult to gather this information for all species of the food web; therefore, accurately estimating the strength of the interactions is infeasible within the scope of this study. Based on the trophic information and network theory, we performed species removal by degree and by trophic level, considering four different proposed thresholds for secondary extinctions, i.e., the proportion of prey that a consumer must lose to become extinct. We examined the impact of extinction simulations on the following network properties: connectance (C), modularity (M) and stability (Quasi-Sign Stability, QSS). Finally, we analyzed cumulative secondary extinctions vs. primary extinctions and evaluated the dependency of M and QSS on food web connectance.

## 2. Materials and Methods

### 2.1. Study Site

Potter Cove (62°14′ S, 58°40′ W) is a tributary inlet of 25 de Mayo/King George Island, Antarctica, and one of the largest fjords of the central South Shetland Islands, measuring 4 km long and 2.5 km wide (Figure 1).

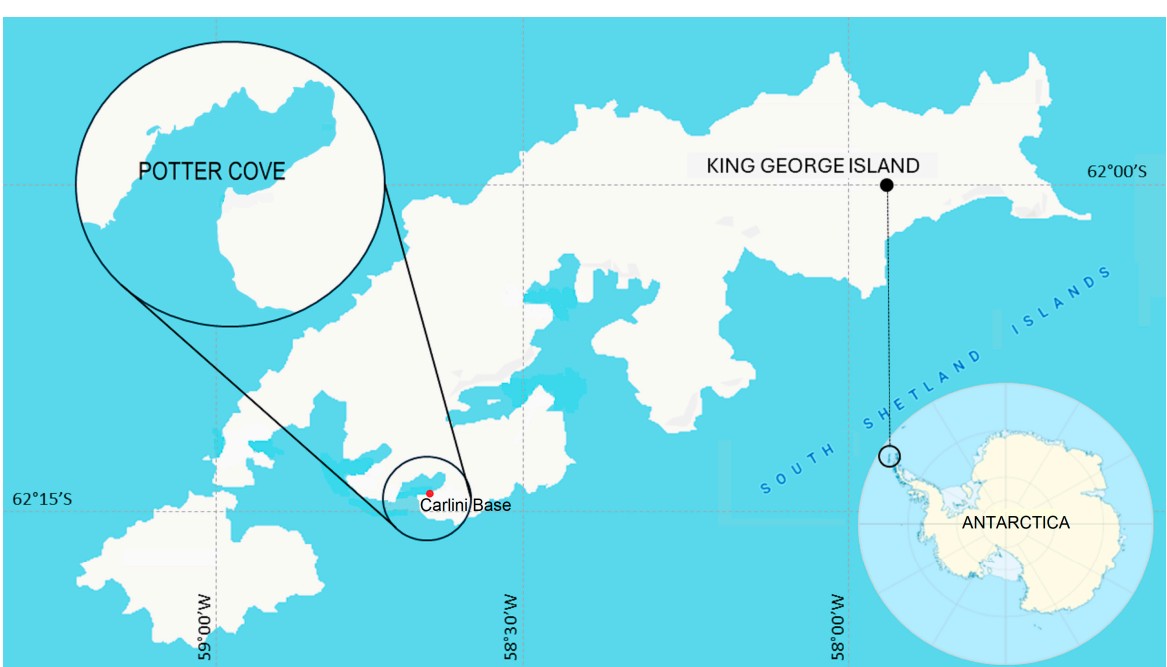

**Figure 1.** Map of Potter Cove, 25 de Mayo/King George Is.; the Argentinian scientific station, Carlini Base, is highlighted.

### 2.2. Food Web Data Set

The Potter Cove food web, first described by Marina et al. [53], was assembled based on trophic information about benthic and pelagic species inhabiting the fjord. This highly resolved food web includes 91 trophic species (nodes), defined as a group of taxa collapsed into a single node in the food web, and 307 feeding interactions (links). Although most of the trophic species were identified at the species level (e.g., *Euphausia superba*), some were defined at a lower taxonomic level due to the lack of detailed trophic information (e.g., ascidians) or when species share the same set of predators and prey (e.g., copepods) [54,55]. Henceforth, "trophic species" will be called "species". More details on the Potter Cove food web assembly process can be found in Marina et al. [53].

### 2.3. Extinction Simulations

Species were removed from the network based on two removal criteria: (1) degree (total number of trophic interactions) where species were removed from most-connected to least-connected species (Descending) and vice versa (Ascending), and (2) trophic level (position of species in the food web with respect to the source of matter and energy), where species were removed from high-trophic level to low-trophic level (High) and vice versa (Low). In addition, we established an extinction sequence considering mid-trophic level species ($2.5 \leq TL \leq 3.5$) and performed it in both ascending (MidAsc: from 2.5 to 3.5) and descending (MidDes: from 3.5 to 2.5) orders. The last criterion was studied in order to understand how the removal of basal, intermediate and top trophic-level species affect the structure and processes of the ecosystem. When two or more species had the same trophic level, we considered a descending criterion by degree, meaning that among species with equal trophic levels, the one with the highest degree was deleted first. Species elimination was carried out until the network was reduced by 90% of its original size (91 species) except for the mid-trophic level extinction sequence, where a fixed number of species were eliminated. Species degree and trophic level can be found in the Appendix A (Table A1).

### 2.4. Thresholds on Secondary Extinctions

For each extinction step, we calculated the number of species lost considering both primary (eliminated species) and secondary extinctions. It has been proposed that a

secondary extinction occurs when one consumer species loses all its prey species [11]. This definition is based on a topological approach, which solely requires the network structure as input, simplifying its application to complex networks. However, this approach presents two limitations: (1) that a secondary extinction occurs "only" if the consumer loses all its prey due to primary extinctions and (2) that all species have the same baseline probability of extinctions, even though in natural systems, some species are more vulnerable than others [20,56]. In order to deal with such limitations, we considered a wide range (0.2–0.8) of secondary extinction thresholds. Following Bellingeri and Bodini [19], we applied different thresholds based on the proportion of prey that a consumer loses. For example, a threshold of 0.2 indicates that a consumer becomes extinct when it loses 80% of its prey, representing the 20% of the original incoming items. In this regard, we included four thresholds: 0.2, 0.4, 0.6 and 0.8.

*2.5. Effect on Food Web Properties*

In order to measure the impact of extinction simulations on the food web, we considered the following properties: connectance (C), modularity (M) and stability (Quasi-Sign Stability). Connectance (C) is a standard food web metric representing the proportion of possible links that are actually realized. It is considered an estimator of community sensitivity to perturbations that strongly covaries with many network properties [12,22,57–59]. Modularity (M) measures how strongly sub-groups (modules) of species interact compared with the strength of interaction with other sub-groups. It is related to network persistence, since the impacts of a perturbation are retained within modules, minimizing impacts on the network [60–62]; values closer to 1 indicate more persistence. The stability of the food web was measured using the Quasi-Sign Stability index (QSS), that is, the proportion of community matrices that are locally stable preserving the sign structure [63]. We calculated QSS considering the mean of the maximum eigenvalue of the random community matrices for easier analysis and visualization; values closer to zero indicate a more stable food web. This index is directly related to network local stability, representing the "return time" the community needs to return to the original equilibrium after a sufficiently small disturbance [64–66]. We calculated C, M and QSS along the extinction sequences, analyzed cumulative secondary extinctions vs. primary extinctions and studied the dependency of properties on connectance.

In order to enhance the robustness and reliability of our results, we used two different Integrated Development Environments (IDE): RStudio and MatLab. Most network metrics were calculated in R version 4.2.2 (R Core Team 2022, Vienna, Austria), using "igraph" [67] and "multiweb" [68] packages. Matlab (R2020a) code was used to set the network for each extinction criteria (i.e., ordering the matrix) and identify the species in each extinction step. Source codes and data are available in the Supplementary Material.

**3. Results**

*3.1. Effects on Connectance (C)*

We observed two expectable trends of C when extinctions were performed by degree: (1) in ascending order, C increased along the extinction sequence, reaching values six times higher than the original value (0.037)—the higher the threshold, the lower the increase— and (2) in descending order, C values decreased abruptly, approaching zero—the higher the threshold, the faster the decrease (Figure 2).

When high-trophic-level species were removed sequentially (High criterion), C values barely varied from their initial value (0.037) until the twentieth step, approximately, where it plummeted by half rapidly. The same trend was observed in subsequent steps (>20). This trend did not change among the thresholds. When low-trophic-level species were removed (Low criterion), C displayed an increasing trend that accelerated at high thresholds ($\geq$0.6). In the case of the MidAsc criterion, two opposite trends were observed: a monotonous slight increase in the first 15 steps, approximately, and thereafter a decreasing trend, where the high thresholds drove a faster decrease. In the case of the MidDes criterion, C showed

a trend similar to that observed when high-trophic-level species were removed (High criterion). Here, the collapse was approximately between steps 15 and 17 (Figure 2).

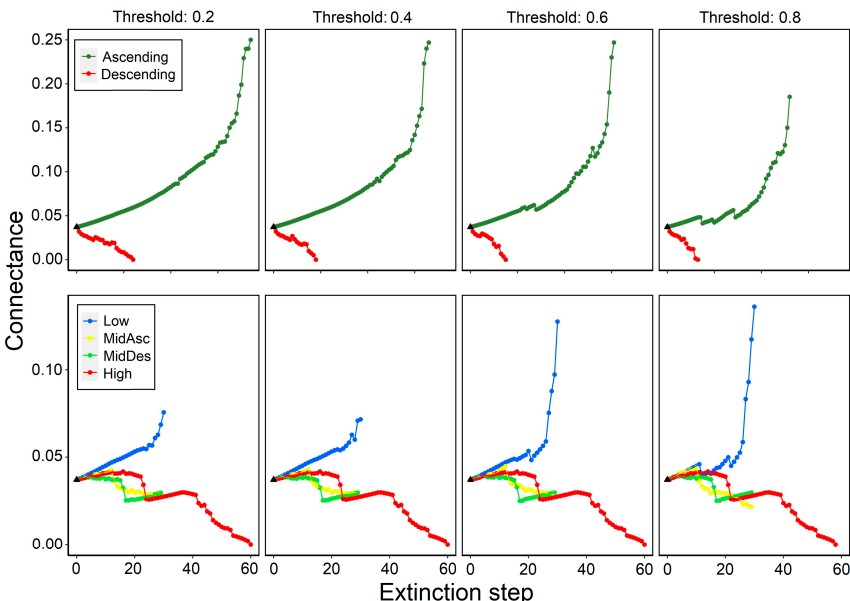

**Figure 2.** Connectance changes when removing species by degree (**top panels**) and trophic level (**bottom panels**), considering four thresholds for secondary extinction (0.2, 0.4, 0.6 and 0.8). Black triangle indicates the original connectance (0.037).

### 3.2. Effects on Modularity (M)

We observed that modularity presented opposite trends regarding species degree deletion: when most-connected species were removed (Descending), M increased during the first 15–20 steps and then it rapidly plummeted; when least-connected species were removed (Ascending), M decreased displaying oscillations and decreased faster at high thresholds ($\geq 0.6$) (Figure 3).

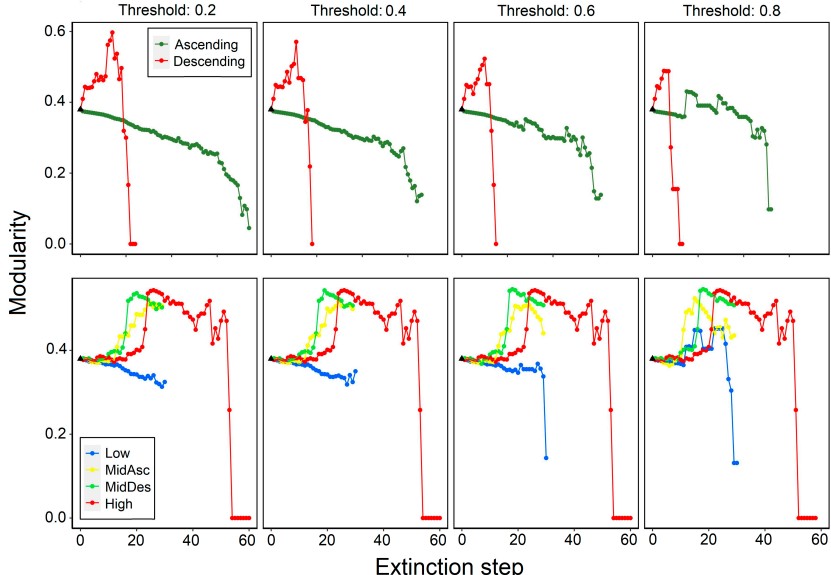

**Figure 3.** Modularity changes when removing species by degree (**top panels**) and trophic level (**bottom panels**), considering four thresholds for secondary extinction (0.2, 0.4, 0.6 and 0.8). Black triangle indicates the original modularity (0.37).

For trophic level extinction criteria, we observed that M increased at the beginning and then slightly decreased, except for the Low criterion, which remained unaltered until a high threshold ($\geq$0.6), at which point it decreased. There was a slight difference between the MidDes and MidAsc criteria, the former presenting a sudden increase in modularity, similar to the High criterion, near the 20th step. No variation in the trend of M for the High criterion, and no significant variation for the MidDes criterion were observed among the thresholds. The Low and MidAsc criteria displayed variations at high thresholds ($\geq$0.6) (Figure 3).

### 3.3. Effects on Stability (QSS)

The QSS decreased rapidly when deleting from most- to least-connected species (Descending). When deleting in Ascending order, it decreased with a slight variation depending on the threshold: the higher the threshold, the faster the decrease. When trophic level was considered, it decreased for all criteria and thresholds. Except for the Low criterion, QSS started to rapidly decrease around the 20th step. There were no changes among thresholds for the High and MidDes criteria (Figure 4).

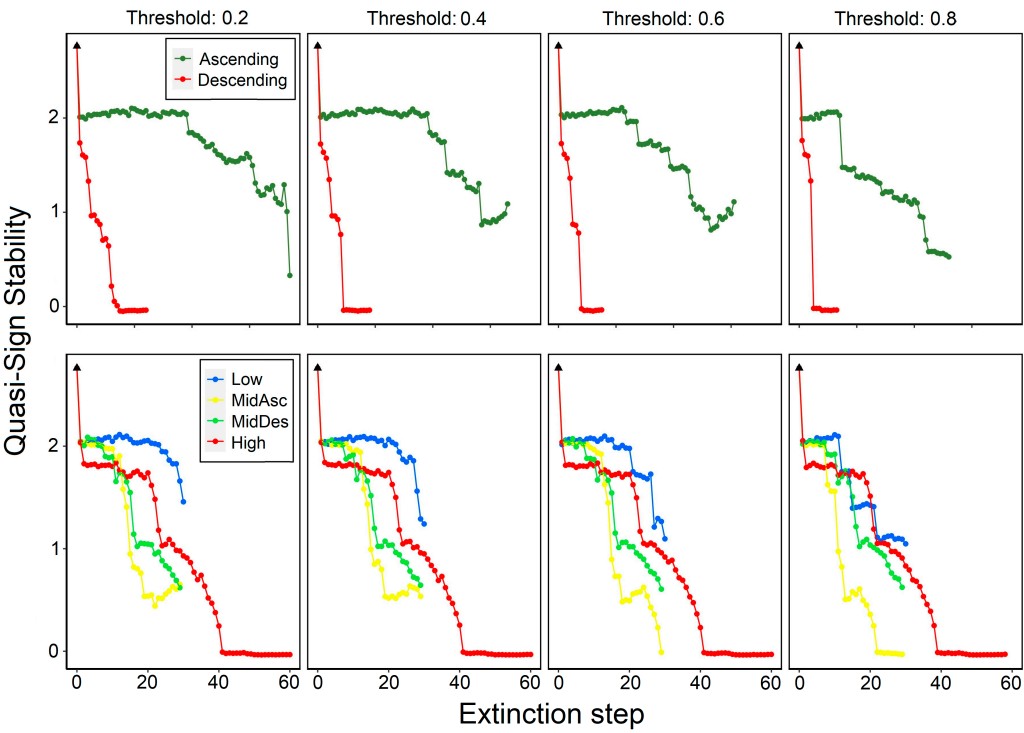

**Figure 4.** Quasi-Sign Stability changes when removing species by degree (**top panels**) and trophic level (**bottom panels**), considering four thresholds for secondary extinction (0.2, 0.4, 0.6 and 0.8). Black triangle indicates the original QSS (2.76).

### 3.4. Cumulative Secondary Extinctions

When most-connected species (Descending) were removed, we observed that the number of secondary extinctions rapidly increased in the first steps (<23), regardless of the threshold. On the contrary, when least-connected species (Ascending) were removed, the number of secondary extinctions varied among the thresholds: low thresholds ($\leq$0.4) displayed a relatively low number of secondary extinctions, while high thresholds ($\geq$0.6) displayed a relatively high number of secondary extinctions (Figure 5).

Removing species by trophic level revealed that the number of secondary extinctions varied for each criterion and threshold, except for the High criterion, where no secondary extinctions were registered before the 0.8 threshold. The highest number of secondary extinctions were observed for the Low criterion followed by MidAsc criterion, where the

higher the threshold, the higher the number of secondary extinctions. There were no changes for the MidDes criterion among the thresholds (Figure 5).

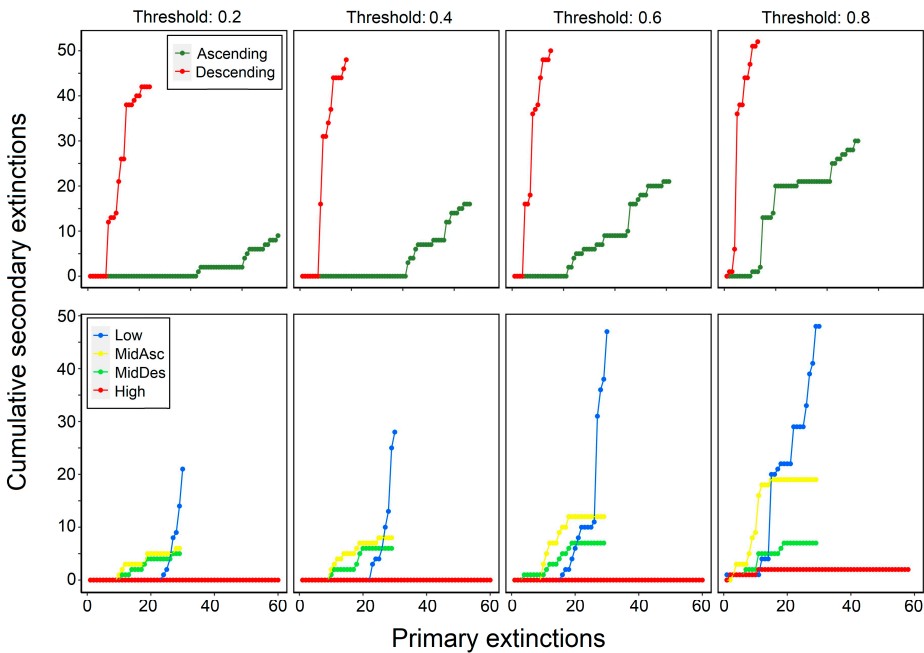

**Figure 5.** Cumulative secondary extinctions versus primary extinctions when removing species by degree (**top panels**) and trophic level (**bottom panels**), considering four thresholds for secondary extinction (0.2, 0.4, 0.6 and 0.8).

*3.5. Properties Dependency on Food Web Connectance*

3.5.1. Modularity

We observed that M decreases when C increases for all thresholds and criteria, except when most-connected (Descending) and high-trophic-level (High) species were removed, in which case M displayed an increasing trend for low values of C (<0.037) (Figure 6).

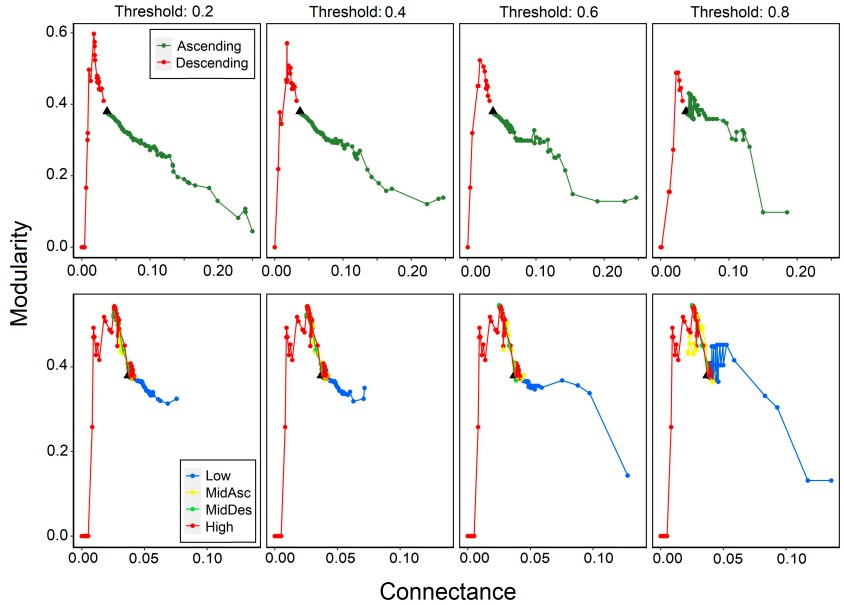

**Figure 6.** Modularity versus connectance when removing species by degree (**top panels**) and trophic level (**bottom panels**), considering four thresholds on secondary extinction (0.2, 0.4, 0.6 and 0.8). Black triangle indicates the original connectance (0.04).

3.5.2. Quasi-Sign Stability

We observed different QSS trends for the different extinction criteria. On the one hand, when most-connected species (Descending) were removed, it rapidly increased for low values of C (<0.037); when least-connected species (Ascending) were removed, it softly decreased for values of C higher than 0.037. On the other hand, when species were removed by trophic level, the QSS displayed a fluctuating and increasing trend for low values of C (<0.037) for all criteria except the Low criterion, for which the QSS displayed a fluctuating and decreasing trend for C > 0.037 (Figure 7).

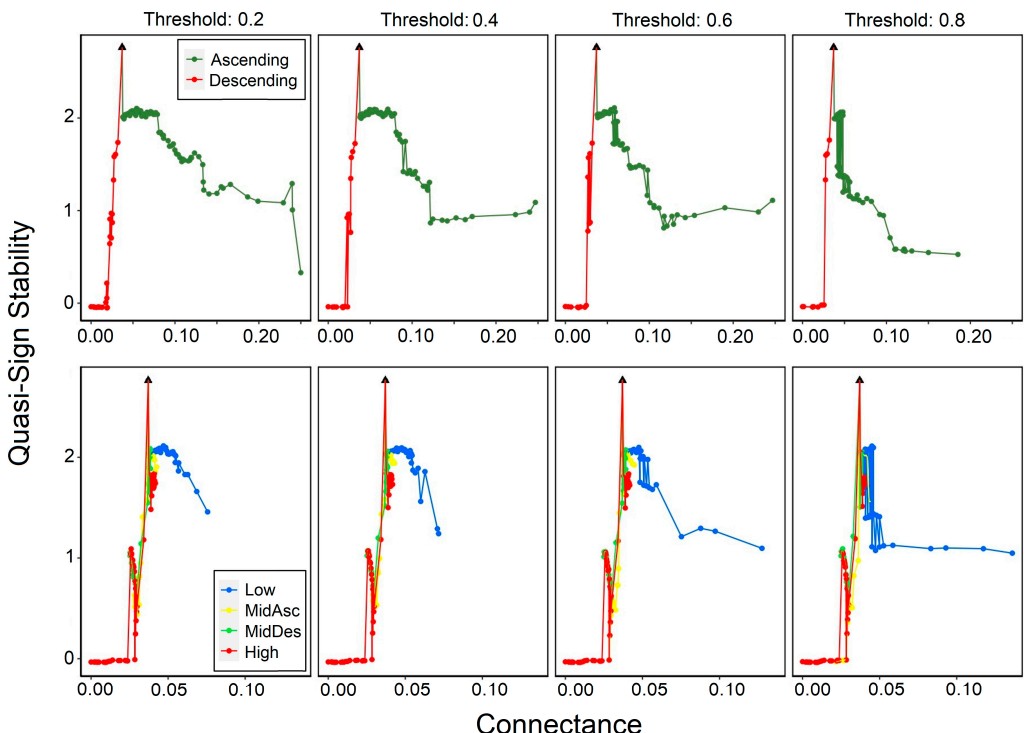

**Figure 7.** Quasi-Sign Stability versus connectance when removing species by degree (**top panels**) and trophic level (**bottom panels**), considering four thresholds on secondary extinction (0.2, 0.4, 0.6 and 0.8). Black triangle indicates the original connectance (0.037).

**4. Discussion**

*4.1. Topological Role of Species*

It is widely known that not all species have the same role in a community (i.e., they make different contributions to the structure and processes), so removing some of them could lead to additional species loss, causing dramatic changes (e.g., trophic cascade) for the whole ecosystem [69–71]. These species, also called "keystone species" [72,73], play critical roles in ecological communities and ecosystem function by interacting directly or indirectly with other species [74], hence, the need for in-depth network analysis to quantify their importance in an ecosystem [75]. In this regard, keystone species are defined by the position they occupy within its community [76], which is ultimately related to properties such as degree (feeding links in which species participate) and trophic level (species vertical position in the food web), since they are good descriptors of the species' potential to affect the rest of its community [77].

Rapid warming along the Antarctic Peninsula has led to the massive loss of ice shelves and retreating glaciers, which has increased sedimentation in coastal areas, exposing marine biota to multiple stresses [78]. Faced with such changes, species may respond by changing their phenology or distribution to follow changing environments; if they cannot do either, they will remain in isolated pockets of unchanged environments or, more likely, disappear. In this regard, Potter Cove is being affected on multiple different scales, from the individual

level to the ecosystem level. Sahade et al. [48] have reported a marked shift from a filter feeder–ascidian domination to a mixed assemblage. Also, changes in the structures of phytoplankton, microphytobenthic and bacterial communities have been shown [79,80], as well as a decrease in primary production, with the potential to steer the seafloor ecosystem towards net heterotrophy [81,82]. Furthermore, changes in species distribution can lead to biological invasions of new habitats in the Antarctic Peninsula [83]. Some studies suggested that ecosystems with low connectance, like Potter Cove, are more vulnerable to invasions than those with high connectance [84,85]. This result is consistent with two important findings concerning invasive species in the region: the king crab (Lithodidae) on the continental slope of the Antarctic Peninsula [86] and the bivalve *Mytilus* sp., reported in the South Shetland Islands [87]. These invasions can lead to profound implications for communities. Populations of king crab generalists have the potential to drastically reshape the structure of marine Antarctic food webs by increasing the number of interactions, thus affecting their connectance and reducing food web modularity [88], consequently impacting their persistence. The presence of the mussel might reduce biodiversity by displacing native species [89,90], implying changes in the complexity and structure of the food web. Certainly, marine Antarctic ecosystems are exposed to many changes and the responses of keystone species to these changes could trigger dramatic effects on the communities.

In Potter Cove, three biological species were identified as potential keystone species: the demersal fish *Notothenia coriiceps*, the brittle star *Ophionotus victoriae* and the amphipod *Bovallia gigantea*. Independently of the extinction criteria used (by degree or trophic level), the removal of these species displayed cascade effects on the connectance, modularity and Quasi-Sign Stability index. On the one hand, these effects can be explained by a combination of species properties where degree and trophic level have major prominence. Regarding the number of interactions, the three species are relatively high-connected: *N. coriiceps* (49), *O. victoriae* (33) and *B. gigantea* (18). The demersal fish and the brittle star are the two most-connected species in the food web and the most important predators due to their omnivorous diets. The fish *N. coriiceps* is a top predator, i.e., it has no predators due to the limits of the food web assembly that did not consider marine mammals or seabirds [53]. The brittle star *O. victoriae* has only two predators in the food web. The amphipod *B. gigantea* has a degree that places it as the fourth most-connected species with only five predators in the food web. The case of the third most-connected species (*Gondogeneia antarctica* (20)) will be discussed later in this section. Regarding trophic level, the three species present a similar and relatively high trophic level in regard to the mean trophic level (2.1): *N. coriiceps* (2.80), *O. victoriae* (2.97) and *B. gigantea* (3.00). Although trophic level alone is not a strong predictor of cascading effect, it is widely known that major predators and primary producers are expected to have particularly large effects on the rest of their communities through top-down and bottom-up control, respectively [77,91,92]. In the food web of Potter Cove, *N. coriiceps* and *O. victoriae* are high-connected species and major predators, which can explain the fact that their removal causes cascading effects on the network properties, suggesting top-down control. On the contrary, although *B. gigantea* has a similar trophic position to *N. coriiceps* and *O. victoriae*, its degree is comparatively low; in fact, there is another amphipod with a higher degree than *B. gigantea*: the above-mentioned *G. antarctica*. The case of these two species and their roles in the food web deserves to be addressed. Even though *G. antarctica* is more connected in the food web than *B. gigantea*, the effect of its removal on network properties was lower. This result might be explained by the fact that: (1) their trophic levels of 2 and 3, respectively, are ultimately linked to their diet (i.e., herbivorous vs. carnivorous), highlighting the importance of trophic position in the food web over number of interactions (when this number is not higher with respect to the mean species degree); and (2) because there are coexisting species that are trophically-equivalent to *G. antarctica* and replace the energy paths when it is removed, which does not occur in the case of top predators from a trophic guild, such as *B. gigantea*. On the other hand, another way to explain the results of removing these three species from the food web could

be through the extent to which species interact with different modules (hyperconnected groups) within the network. A recent study that compared the stability of sub-Antarctic and Antarctic food webs discussed the topological role of the species of the Potter Cove food web [93]. They found that *N. coriiceps* is the only species with high connectivity between and within modules, playing the role of "network connector", while *B. gigantea* and *O. victoriae* were identified as "module connectors" or species whose interactions are mostly between modules. Thus, they suggested that *N. coriiceps* represents the most important species responsible for linking modules and connecting the entire food web.

### 4.2. Effects of Thresholds on Food Web Properties

Biodiversity loss simulations have been extensively conducted by many studies in the last 30 years for different ecosystems [10–18]. Some of these studies have included the biological data of species (e.g., biomass and body size) in order to measure interaction strength and develop dynamical models [15,17], and only a few have considered thresholds for secondary extinctions in order to assess biodiversity loss based on topological approaches [19,20]. Furthermore, how these thresholds impact food web properties beyond considering species degree as the only extinction scenario has not been addressed so far. Here, we used a topological approach to extinctions due to the lack of biological data for each trophic species of the Potter Cove food web. Certainly, dynamical analysis is a powerful tool for predicting secondary extinctions and detecting indirect effects, such as top-down extinctions caused by the loss of a top predator [20]. However, the construction of dynamic food web models necessitates the specification of numerous parameters, interaction strengths between species and functional groups (often characterized by high uncertainty and considerable costs for resolution in real systems). In this study, the absence of information regarding the strength of the interactions constitutes a limitation; however, it is of considerable importance to note that a topological approach reduces dependence on extensive systems knowledge, requiring only information about the structure of the network and enabling the analysis of more complex food webs [18]. Furthermore, it has been demonstrated that the topology of the network—specifically, the presence or absence of interactions—plays a decisive role in the local stability of the food web, exerting a more substantial influence on stability than the interaction forces themselves [63].

In the Potter Cove marine food web, the inclusion of thresholds to assess secondary extinctions in the different scenarios (e.g., the removal of species by trophic level) resulted in different patterns. We observed that the effects of biodiversity loss on food web properties (i.e., connectance, modularity and Quasi-Sign Stability index) displayed similar trends among thresholds in most of the scenarios considered, except for those where basal or low-trophic-level species were the first species removed from the network. These results might be related to the high number of basal and low-trophic-level species in the food web (see Marina et al. [53]) that constitute the food items of most consumers in the network (90% of consumers are connected to at least one basal species). Then, increasing consumers' sensitivity to the loss of their primary resources (i.e., the consumer becoming extinct after a certain fraction of its prey were removed) implies the highest impact on food web properties among the thresholds compared with other extinction scenarios; increasing food web sensitivity that could suggest a "bottom-up" control. In the last few decades, many studies have documented the importance of primary producers for consumers in polar ecosystems [94–96], especially in Antarctic communities where micro- and macroalgae have been regarded as primary food sources, providing an important proportion of carbon to Antarctic benthic consumers [97–99]. In Potter Cove, the role of macroalgae in the regulation of the food web has been studied recently by performing in silico experiments and considering different bottom types [20,100], showing that no cascading effects were observed in macroalgae extinction events until a high threshold was reached. This fact highlights the sensitivity of the food web to the elimination of other basal species (e.g., phytoplankton), which can cause a trophic cascade. Furthermore, Cordone et al. [20,100] have shown that the Potter Cove ecosystem displays a robust response against environmental

perturbations, which can be linked to food web redundancy. In Potter Cove, biodiversity loss experiments showed that the removal of most of the species does not have a cascading effect on the ecosystem, which could be related to species having similar functional roles in a community, since many of the species in Potter Cove are similar in their trophic [20] and non-trophic [101] interactions, and their loss is not critical to the ecosystem. Thus, species with similar roles may be able to compensate if one species becomes extinct, increasing food web resistance by means of the availability of alternative prey [69,102]. Finally, due the complexity of this ecosystem, where species establish different types of interactions (i.e., trophic, commensalism and mutualism) in an intricate manner, deciding whether the regulation of numbers or biomass is primarily controlled by bottom-up, top-down or wasp-waist effects remains a challenge [25].

### 4.3. Multidimensional Stability Criteria

Disentangling the complexity–stability relationship of food webs is a challenge that sparks interest and debates among environmentalists even today [103]. We understand stability as a complex concept that can be analyzed in numerous ways, which is one of the things that motivated this study. In this study, stability was discussed under the assumption that Potter Cove is in a steady state, considering complexity and structural properties: connectance, modularity and Quasi-Sign Stability index. Connectance—directly linked to food web complexity and probably the most-studied property in the last three decades [104]— displayed a high impact when most-connected species were removed (i.e., cascading effect), which is consistent with many previous works that suggested that food web sensitivity to perturbations is ruled by the number of species and their degree [11–13,26,40,41]. As we discussed above, the degree and interaction partners of species play a fundamental role when analyzing food web persistence against biodiversity loss. Depending on the distribution of interactions, some species provide more interactions between modules than within modules, so it would be expected that their elimination increases module robustness, reducing the propagation of perturbations and therefore increasing network stability [105]. In the Potter Cove food web, the highest values of modularity were reached when the aforementioned keystone species (*N. coriiceps*, *O. victoriae* and *B. gigantea*) were removed, which is consistent with their topological role, i.e., network and module connectors. Furthermore, the loss of these species displayed similar effects on network stability. We observed a rapid change in stability along the extinction sequences, which suggests a transition from less to more stable state. This result might be explained by the relationship between complexity and stability: the lower the complexity, the higher the stability.

Potter Cove's complexity and structure were initially studied by Marina et al. [53], who suggested the potential fragility of the food web due to its low values of connectance; however, subsequent studies have found the network relatively robust against disturbances [20,93,100]. This apparent contradiction highlighted the need for a deeper understanding of ecosystem complexity by considering the types of species interactions and the relationships between species functional roles within the ecosystem's structure [25,101]. Taking these suggestions into account, we extended previous findings by analyzing the complexity–stability relationship of the Potter Cove food web through the correlation between modularity, stability and connectance. While connectance provides information about food web complexity, modularity and the Quasi-Sign Stability index bring out an idea about global and local stability of the ecosystem, respectively. This method revealed two types of stability regimes around food web complexity. On the one hand, we found relatively high values of modularity around the original connectance, where the lower the connectance, the higher the modularity, which suggested high persistence against perturbations. On the other hand, we observed an abrupt change on the Quasi-Sign Stability index in a close vicinity to the original connectance: a small variation in the values of connectance caused a large decrease of this index, bringing it closer to zero and suggesting a high sensitivity to small disturbances in complexity. In general, this behavior was observed in both modularity and Quasi-Sign Stability regardless of the extinction thresholds and criteria

used. The apparent contradiction previously mentioned may be explained by suggesting that Potter Cove could be locally unstable in its initial state due to its sensitivity to small perturbations but globally stable against long-range perturbations.

**Author Contributions:** Conceptualization, V.S., G.C. and T.I.M.; methodology, V.S. and T.I.M.; software, V.S., G.C. and T.I.M.; validation, V.S. and G.C.; formal analysis, V.S., G.C. and T.I.M.; investigation, V.S.; writing—original draft preparation, V.S.; writing—review and editing, G.C., T.I.M. and F.R.M.; visualization, T.I.M.; supervision, F.R.M.; project administration, V.S.; funding acquisition, V.S. All authors have read and agreed to the published version of the manuscript.

**Funding:** This research was funded by the CoastCarb (coastal ecosystem carbon balance in times of rapid glacier melt) international Research Network funded by the Marie Curie Action RISE (Research and Innovation Staff Exchange) of the Horizon 2020 Framework Programme of the European Union (H2020-MCSA-RISE 872690), and the Universidad Nacional de General Sarmiento (UNGS Res. N° 209/98).

**Institutional Review Board Statement:** Not applicable.

**Data Availability Statement:** The original contributions presented in the study are included in the article, further inquiries can be directed to the corresponding author. Source codes and data about extinction simulations are available in the public GitHub repository (https://github.com/vasalinas/Extinction-simulations, accessed on 14 December 2023).

**Acknowledgments:** This work was conducted in the frames of VS's secondment related to the RISE project CoastCarb, endorsed by the Universidad Nacional de General Sarmiento (UNGS, Argentina).

**Conflicts of Interest:** The authors declare no conflicts of interest. The funders had no role in the design of the study; in the collection, analyses or interpretation of data; in the writing of the manuscript; or in the decision to publish the results.

## Appendix A

**Table A1.** Trophic level (TL) and Degree of each species of Potter Cove food web.

| Species | TL | Degree |
| --- | --- | --- |
| *Urticinopsis antartica* | 4.27 | 4 |
| Octopus | 4.13 | 4 |
| *Chaenocephalus aceratus* | 4.02 | 4 |
| *Protomyctophum* | 3.70 | 1 |
| *Diplasterias brucei* | 3.67 | 1 |
| *Trematomus newnesi* | 3.65 | 10 |
| *Trematomus bernacchi* | 3.59 | 7 |
| *Parachaenichthys charcoti* | 3.50 | 1 |
| *Perknaster fuscus antarticus* | 3.46 | 4 |
| *Parborlasia corrugatus* | 3.41 | 9 |
| *Odontaster meridionalis* | 3.35 | 7 |
| Hyperiids | 3.33 | 6 |
| *Harpagifer antarcticus* | 3.32 | 11 |
| *Notothenia rossii* | 3.25 | 8 |
| *Margarella antarctica* | 3.25 | 10 |
| *Perknaster aurorae* | 3.25 | 2 |
| *Sterechinus neumayeri* | 3.21 | 17 |

**Table A1.** *Cont.*

| Species | TL | Degree |
|---|---|---|
| *Glyptonotus antarcticus* | 3.13 | 8 |
| *Lepidonotothen nudifrons* | 3.07 | 7 |
| *Austrodoris kerguelensis* | 3.07 | 10 |
| *Odontaster validus* | 3.06 | 10 |
| *Bovallia gigantea* | 3.00 | 18 |
| *Ophionotus victoriae* | 2.97 | 33 |
| *Notothenia coriiceps* | 2.80 | 49 |
| Salps | 2.70 | 8 |
| *Neobuccinum eatoni* | 2.67 | 11 |
| *Dacrydyum* sp. | 2.50 | 3 |
| *Euphausia superba* | 2.50 | 11 |
| Copepods | 2.50 | 5 |
| Ascidians | 2.50 | 5 |
| Oligochaetes | 2.50 | 3 |
| Hydrozoans | 2.50 | 4 |
| Bryozoans | 2.50 | 5 |
| Priapulids | 2.50 | 2 |
| Mysids | 2.50 | 3 |
| *Malacobelmnon daytoni* | 2.50 | 2 |
| *Laternulla elliptica* | 2.33 | 6 |
| *Haliclonidae* sp. | 2.25 | 11 |
| Stylo-Myca | 2.25 | 13 |
| *Rosella* sp. | 2.25 | 11 |
| *Dendrilla antarctica* | 2.25 | 6 |
| *Nereidae* | 2.00 | 17 |
| *Eatoniella* sp. | 2.00 | 7 |
| *Nacella concinna* | 2.00 | 9 |
| *Laevilacunaria antarctica* | 2.00 | 9 |
| *Paradexamine* sp. | 2.00 | 7 |
| *Eurymera monticulosa* | 2.00 | 9 |
| *Pontogeneiella* sp. | 2.00 | 8 |
| *Gondogeneia antarctica* | 2.00 | 20 |
| *Pariphimedia integricauda* | 2.00 | 3 |
| *Cheirimedon femoratus* | 2.00 | 4 |
| *Gitanopsis antarctica* | 2.00 | 5 |
| *Prostebbingia gracilis* | 2.00 | 14 |
| *Waldeckia obesa* | 2.00 | 6 |
| Hippo-Orcho | 2.00 | 3 |
| *Oradarea bidentata* | 2.00 | 3 |
| *Serolis* sp. | 2.00 | 3 |
| *Plakarthrium puncattissimum* | 2.00 | 5 |

**Table A1.** *Cont.*

| Species | TL | Degree |
|---|---|---|
| *Hemiarthrum setulosum* | 2.00 | 3 |
| Zooplankton | 2.00 | 17 |
| *Callophyllis atrosanguinea* | 1.00 | 1 |
| *Curdiea racovitzae* | 1.00 | 3 |
| *Georgiella confluens* | 1.00 | 3 |
| *Gigartina skottsbergii* | 1.00 | 5 |
| *Iridaea cordata* | 1.00 | 5 |
| *Myriogramme manginii* | 1.00 | 2 |
| *Neuroglossum delesseriae* | 1.00 | 1 |
| *Palmaria decipiens* | 1.00 | 9 |
| *Pantoneura plocamioides* | 1.00 | 1 |
| *Picconiella plumosa* | 1.00 | 1 |
| *Plocamium cartilagineum* | 1.00 | 4 |
| *Pyropia plocamiestris* | 1.00 | 1 |
| *Trematocarpus antarcticus* | 1.00 | 1 |
| *Adenocystis utricularis* | 1.00 | 3 |
| *Ascoseira mirabilis* | 1.00 | 3 |
| *Desmarestia anceps* | 1.00 | 2 |
| *Desmarestia antarctica* | 1.00 | 3 |
| *Desmarestia menziesii* | 1.00 | 5 |
| *Geminocarpus geminatus* | 1.00 | 2 |
| *Phaeurus antarcticus* | 1.00 | 3 |
| *Lambia antarctica* | 1.00 | 1 |
| *Monostroma hariotii* | 1.00 | 3 |
| *Urospora penicilliformis* | 1.00 | 1 |
| *Ulothrix* sp. | 1.00 | 1 |
| Epiphytes diatoms | 1.00 | 8 |
| Benthic diatoms | 1.00 | 15 |
| Phytoplankton | 1.00 | 16 |
| Aged detritus | 1.00 | 5 |
| Squids | 1.00 | 3 |
| Fresh detritus | 1.00 | 12 |
| Necromass | 1.00 | 9 |

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
