# Peer review of "Estimating the Impact of Biodiversity Loss in a Marine Antarctic Food Web"

_diversity, doi:10.3390/d16010063_

Round 1
Reviewer 1 Report
Comments and Suggestions for Authors
This manuscript examined the impact of extinctions on connectance, modularity, and stability of the marine food web of Caleta Potter (25 de Mayo Island, Antarctica) based on species sequence removal simulating. The authors found different responses of food web properties depending on the extinction criteria used. The study provided insights for identifying the affecting factors on food web stability. I put forward some questions and suggestions to the authors. I suggest that the manuscript be published after revision.
1. This manuscript used degree-sorted species removal criteria. Actually, only considering the degree of species is not sufficient. So, it is recommended to consider the strength of trophic interactions between species. In food web, connection strength is a crucial but frequently disregarded factor.
2. Whether it is necessary to establish an extinction sequence of a degree-trophic level dual factors to analyse the impact on the food web properties;
3. The manuscript focused on keystone species and their ecological effects in the discussion section, so I suggest adding keystone species removal into the result section, which corresponds to the discussion part. It is important for species conservation.
Author Response
Please find attached the response to the reviewer comments.
Sincerely
Vanesa Salinas

Reviewer 2 Report
Comments and Suggestions for Authors
Impact of biodiversity loss on the structure and stability of a marine Antarctic food web
The manuscript is easy to read and interesting. The methods are adequate and allow for reaching the described conclusions. The one negative thing that I must mention concerning this manuscript, is that it refers to a very particular region, not allowing for general, wider, conclusions. But this is not uncommon in food web ecology. There are plenty of studies relying on one or a few food webs. I find particularly interesting the discussion around stability and complexity. The manuscript is interesting and may provide relevant information. I recommend it to be accepted, after minor revisions.
Abstract
The abstract is effective in showing the work done in this manuscript.
Keywords
Ideally, keywords should not be also in the title. I suggest changing the keywords that are also in the title.
Introduction
The introduction is well-structured and adequately frames the study in the current research landscape.
Line 85 - Where the authors write: “…work properties: connectance (C), modularity (M) and stability (Quasi-Sign Stability).” They should add “, QSS” after “Stability”.
Methods
The methods are very well explained and are adequate to the research question.
Results
Regarding the structure of the results, I think the authors would make it easier for the readers to follow the paper if the figures are shown right after being mentioned in the text. Currently, the figures are all grouped at the end of the result.
Discussion
Lines 316-319 – Please clarify this sentence.
Author Response
Please find attached the response.
